# Elderly-friendly indoor vertical dimensional layout method based on joint mobility

**Feng Wang**[1], **Isarachai Buranaut**[2], **Bo Zhang**[1], **Jie Liu**[3]*

1 Shandong Youth University of Political Science, Jinan, 250109, Shandong, China, 2 Silpakorn University, Bangkok, Talingchan, Thailand, 3 Qilu Normal University, Jinan, 250200, Shandong, China

* 20188528@qlnu.edu.cn

**Data Availability Statement:** All relevant data are within the paper and its Supporting Information files.

**Funding:** The author(s) received no specific funding for this work.

## Abstract

### Introduction

The vertical dimensional arrangement of space is primarily influenced by the reachable range of human fingertips. Currently, this dimensional layout analysis concentrates solely on static body dimensions based on Farley's principle. However, the joint mobility of the elderly population has diminished, necessitating the identification of the factors influencing the age-friendly spatial vertical dimension layout through experimentation.

### Methods

A random sample of 62 adults and 62 elderly individuals were selected to measure the joint mobility of 8 groups with varying comfort levels. The measurement results were analyzed using an independent sample t-test with SPSS software.

### Results

The joint mobility of the elderly group exhibited a notable difference from that of the adult group across distinct comfort gradients. A significant reduction in the range of motion of all joints in the elderly was observed. The findings suggest that the reachable range of the upper limbs of the elderly should be considered in conjunction with joint mobility. We propose a vertical dimensional layout method for residential spaces that accommodates the joint mobility of elderly users.

### Conclusion

The joint mobility of the elderly is significantly deteriorating, and the traditional vertical spatial layout method cannot adequately support the daily life of the elderly. It is essential to integrate the joint mobility factor into the vertical dimensional layout design process. In this paper, we propose an elderly-friendly vertical spatial dimension layout method. It offers a reference for subsequent planning of elderly-friendly vertical dimension layout.

**Competing interests:** The authors have declared that no competing interests exist.

## Introduction

It has been forecasted that will increase from 24 million to 418 million within four decades, from 2011 to 2050 [1]. According to international standards, a country or region enters an ageing society when the proportion of its population aged 65 and over exceeds 7 per cent; it enters a deeply ageing society when the ratio of its population aged 65 overreaches 14 per cent; and it enters a super -ageing society when the proportion of its population aged 65 and over exceeds 20 per cent [2]. In China, there are 260 million people aged 60 and above (18.70 per cent of the total population), of whom 190 million are aged 65 and above (13.50 per cent of the total population) [3]. According to the National Bureau of Statistics of China report, the ageing of China's population has further advanced. Moreover, the traditional living pattern of older people living with their children in China has changed, and the number of older people living alone is increasing [4]. The Chinese government proposed to promote home care services comprehensively [5]. With fewer children to look after, the residential indoor spatial environment needs to provide more targeted external support to the elderly living independently.

The ability of older adults to live independently is governed by physical decline. The conventional vertical dimensional working area height analysis is based on Farley's principle. As shown in the Fig 1, the middle point represents the shoulder peak position; the small circle represents the normal vertical dimensional working area when the person is upright; the large circle represents the maximum vertical plane working area when the person is upright. Because people in the vertical plane can take different body postures such as upright, upper limb bending, squatting, kneeling, single leg kneeling, etc., so the vertical plane working area can be extended down to the ground. It can be seen that this principle only considers static body dimensions. Applying this principle to the vertical dimension related to the operating plane height analysis is to consider only the impact of the static physiological dimensions of the human body such as shoulder peak height and limb length. Such an analytical calculation process acquiesces to the normal mobility of all joints in the human body and is applicable for the average adult.

In addition to a significant decrease in height compared to adulthood [6], there is also a tendency for the range of motion of joints to decrease [7]. We note that some studies have demonstrated some reduction in joint range of motion in older adults. Stiffening strategy was used in

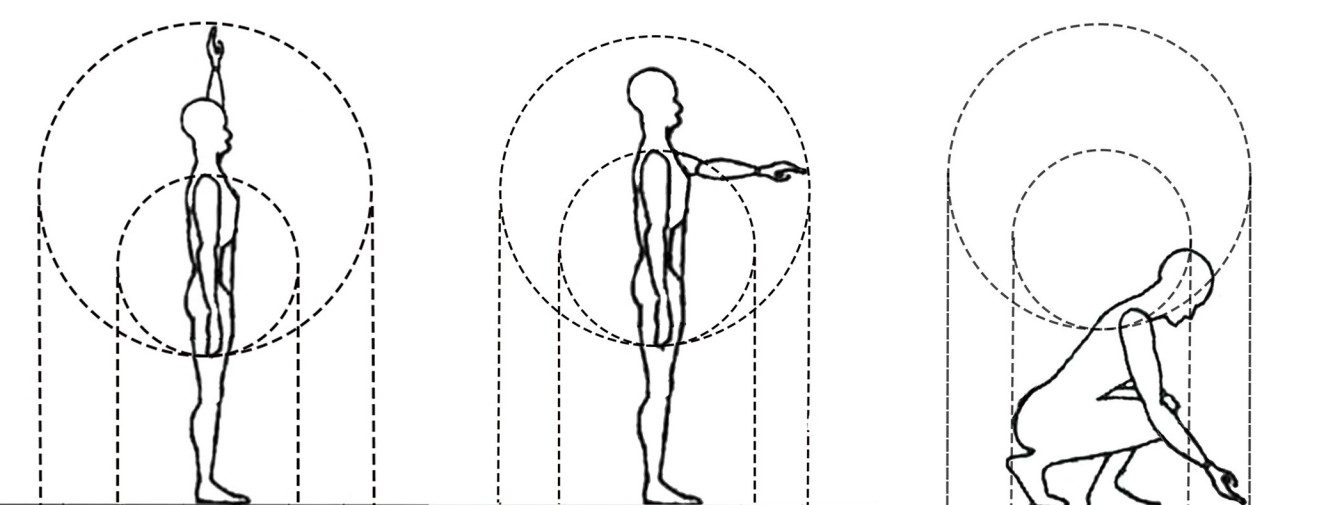

**Fig 1. Schematic diagram of Farley's principle.**

the ankle joint to compensate for the degradation of the postural control system in order to promote body stability and reduce the risk of falling off. Such a strategy reduced the range of motion of the ankle joint in the elderly group [8]. Joint range of motion, also known as joint mobility, refers to the angle of the arc of motion through which the joint moves [9]. Joint mobility is mainly limited by the bones, muscles, and connecting tissues around the joint [10]. The earliest studies by scholars began with the development of the initial joint mobility measures [11]. Subsequently, some scholars began to study joint mobility in older adults and found that the decline in joint mobility was mainly in the vertical dimension of the joint. In contrast, the angle of joint movement in the horizontal dimension was not significantly different from that of adults [12,13]. Another group of scholars studied joint comfort gradients in adults and older adults, and several teams classified joint movement angles according to three gradients of"neutral, mild, and severe"; such studies showed that both adults and older adults could distinguish between the different comfort gradients of joints [14]. This capability provides a quantitative basis for the spatial stratification of the spatial gradient in housing interiors.

In general, the degenerative changes in the bodies of older people, in addition to the decrease in height, also lead to a significant decline in the mobility of many joints in the vertical dimension, ultimately reducing their fingertip reach in the vertical dimension [15]. The spatial layout of the vertical dimension of an interior is the greatest constraint on the daily independent living of older adults [16]. At the same time, the number and variety of inclusions in the residential spatial environment, as well as the constant introduction and application of new inclusions, make it difficult for interior designers to clearly individually label all inclusions in their specific locations in the vertical dimension of the home.

Also, According to the results of the study, the accessibility range of the vertical dimension of the elderly decreases significantly with age, especially for the lower limb joints. The vertical dimension accessibility calculation model constructed in this study can provide a reliable basis for the vertical dimension layout of indoor space for the elderly. The study also proposes a vertical division system for indoor space, which divides the indoor space into 11 reference layers according to the physical activity ability of the elderly. In addition, the study proposes the use of a comfort zone system to improve the safety and comfort of elderly living environments. Overall, the study provides a valuable reference for the design of elderly-friendly indoor space.

Therefore, this study proposes a systemic approach to the layout of the vertical dimension of the interior of a house by considering the declining joint mobility of elderly users. The method is developed based on the anthropometric measurements of elderly users, which are then used to construct a vertical dimension layout system for them. The results of this study are expected to serve as a guide for the division of the vertical dimensional layouts of residential interior designs for the elderly and to provide a key elevation reference for various inclusions in residential interiors for elderly users, which can be applied in the design of ageing-friendly spaces.

## Materials and methods

### Participants

The sample population was consisting of adults and the elderly. The sample size was determined based on the studies by Nurul Shahida et al [17]. and Hu et al [18]. The formula given in Annex A of ISO 15535: 2003 "General requirements for establishing anthropometric databases" was used to estimate the minimum sample size in this study. The 95% confidence interval was used for the 5th and 95th percentiles since in most cases, the anthropometric data that

are of interest to product designers are those at the 5th and 95th percentiles.

$$n \geq (3.006 \times CV/\alpha)^2,$$

In this formula, n, CV and a represents the sample size, coefficient of variation and percentage of the desired relative accuracy, respectively. It was found that the minimum sample size for this study is 112(Fifty-six people in each age group). The sample size was determined based on the assumption that a relative accuracy of 10% is sufficient for the 5th and 95th percentiles and an empirical CV value of 25. Therefore, a total of 124 subjects (Sixty-two older adults aged 60 years and older versus 62 adults aged 18 to 60 years) would be sufficient for this study.

In this study, a stratified integer sampling design was employed for data extraction, ensuring representation from different constituencies. Stratified sampling was utilized to enhance the precision of the same sample size, while integer sampling was employed to improve both sample size and precision within a specified cost.

One hundred and twenty-four participants participated in the study, Sixty-two older adults aged 65 years or older (50 per cent) and sixty-two adults aged 18 to 65 (50 per cent). The present investigation was conducted in Shandong Province, where all participants were in good health and able to perform most home activities independently and smoothly. The subjects possessed well-functioning joints throughout their bodies, with no significant pathologies detected. They had no history of joint surgery nor any joint replacements. Participants unable to stand independently were excluded from the study. The mean age of the younger group was 69.3 years, and the elderly group's mean age was 41.4 years.

Preceding the start of data collection, subjects were informed that the study sought to develop an anthropometric database to improve the vertical dimensional layout of elderly home environments. The measurement procedure was thoroughly explained, and participants could choose whether to partake in the survey after comprehending the study's purpose and procedures.

## Methods

### Ethics statement

This study was approved by the Academic Committee of the School of Art Design, the Shandong Youth University of Political Science. All the work in this study follows the procedures approved by the committee. All participants provided written informed consent.

### Selection of body anthropometric dimensions

In this study, the anthropometric body dimensions of each participant were measured following the standard procedures outlined in ISO 7250—basic human body measurements for technological design, Part 1: body measurement definitions and landmarks [19]. Thirteen variables were selected from the anthropometric dimensions recommended by Kroemer et al. [20–23]. All the anthropometric measurements are shown in Table 1. These dimensions were chosen because these variables are directly related to the spatial vertical dimensional layout of residential interiors for the elderly [24]. Although anthropometric measurements can be measured from both the right and left sides of the body, this study used only the usual side of the tested person due to limited resources.

### Definition of measurement dimensions

As shown in Fig 2, all measurement dimensions are defined as follows [19–21,23]: 1. Natural standing height: the vertical distance from the point of the crown of the head to the horizontal

**Table 1. Significant anthropometric body dimensions.**

| Dimension number | Measure |
|---|---|
| 1 | Natural standing height |
| 2 | Standing eye height |
| 3 | Natural standing shoulder height |
| 4 | Forearm length |
| 5 | Upper arm length |
| 6 | Torso length |
| 7 | Thigh length |
| 8 | Tibial point height |
| 9 | Elbow joint angle |
| 10 | Shoulder joint angle |
| 11 | Torso angle |
| 12 | Knee angle |
| 13 | Calf angle |

ground; 2. Standing eye height: the sheer length from the pupil to the; 3. Natural standing shoulder height: the vertical distance from the shoulder peak point to the horizontal ground; 4. Forearm length: linear distance from the radius to the of the radial styloid process; 5. Upper arm length: linear distance from the point of the shoulder crest to the point of the radius; 6. Torso length: linear distance from the shoulder to the point; 7. Thigh length: point of the trochanter to the point of the tibia; 8. Tibial point height: with vertical standing of the lower leg, the vertical distance from the of the tibia to the ground; 9. Elbow joint angle: the angle between the of rotation of the elbow and shoulder joints and the line between the of rotation of the elbow and wrist joints; 10. Shoulder joint angle: the angle between the line between the centres of rotation of the elbow and shoulder joints and the line from the node of the greater trochanter of the femur to the node of the of the shoulder; 11. Torso angle: the angle between the line from the greater trochanteric node to the shoulder's central node and the vertical line of the more significant trochanteric node; 12. Knee angle: the angle between the line from the node of the greater trochanter to the centre of the knee and the line from the centre of the knee to the lateral ankle; 13. Calf angle: the angle between the centre of the knee to the lateral ankle line and the horizontal ground in the direction of the foot.

**Measuring equipment.** To ensure optimal precision, reliability, and accuracy of the anthropometric readings [25], the body dimensions were measured by trained staff using validated techniques [11]. The equipment needed for this experiment was a manual anthropometer to measure body dimensions for standing postures [13,26] and a universal goniometer to measure the mobility of the desired joint [27].

**Procedure.** Key feature parameter acquisition: Continuing the parameter selection of other scholars' related studies, the three grades of"Neutral, Mild, and Severe" were used [20]. The angle values corresponding to the five parts of the test subject's elbow, shoulder, torso, knee, and the corresponding calf angle were obtained; the limit method was used to measure the corresponding joint movement angle for each joint at the three grades. A camera was used to record the test procedure, as shown in Fig 3. To ensure the accuracy of each value, all variables were required to be measured twice. However, a third measurement was taken when the difference between the two measurements was more than 10 degrees. This way, it was possible to determine which of the first two measurements was incorrect. When it appeared to be impossible to identify the incorrect measurement because the third measurement was in between the former two assessments, and the difference concerning each of them was less than

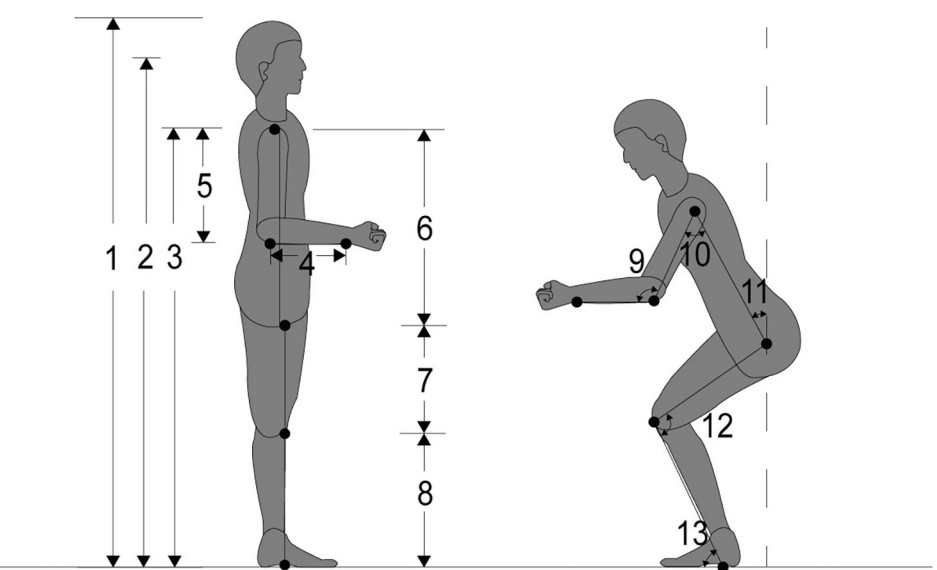

**Fig 2. Measurement dimensions.**

ten degrees, or because the third measurement was again more than 10 degrees more significant/smaller than the largest/smallest measured value, the mean value of all three measurements was calculated. In all other cases, the mean value of two measurements was computed and used as input for the analyses.

## Data collection for "Neutral" grade

The experimenter asked the test subjects to wear daily home clothing and shoes and to maintain a natural standing posture and stationary movements for 10–15 s. The experimenter measured the body dimensions and the angle of each joint. Two sets of measurements were

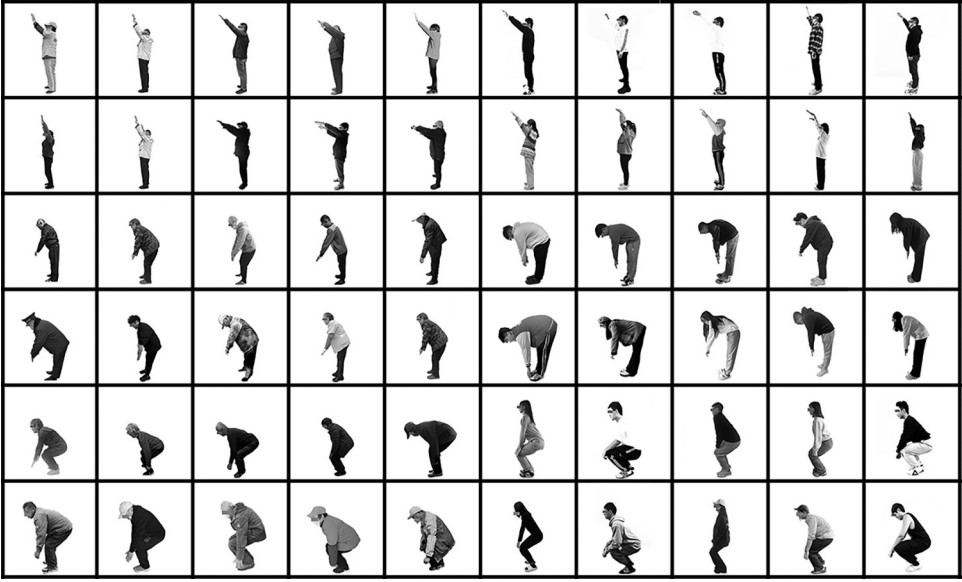

**Fig 3. Test process record (partial).**

required and based on the results of each data collection; it was determined whether a third measurement was required.

## Data collection for "Mild and Severe" grades

The experimenter asked the test subjects to wear daily home clothing and shoes and to maintain a natural standing posture. Then, the experimenter commanded them to bend the knee to the Mild grade angle. After completion of the verbal instruction of the experimenter, while maintaining a resting action for 15s, the experimenter measured the joint angle corresponding to the Mild grade posture of the knee. The experimenter gave the command to bend the knee to the Severe grade angle, and, again, while maintaining a resting action for 15s, the experimenter measured and recorded the joint angle. After completing the task, the body returned to the natural standing posture. Similar commands were given again, such as "bend the elbow to a posture that feels mild" and "bend the shoulder to a posture that feels mild", with elbow, shoulder, torso, and knee movements being requested randomly; the first command could also be to bend to the "Severe" grade, and the second command could be to bend to the "Mild" position. Finally, two sets of eight movements were completed for the "Mild" and "Severe" grades.

**Statistical analysis.** The data were analysed separately for the two groups. A descriptive analysis was performed for the elbow joint, shoulder joint, torso angle, knee joint, and their corresponding eight categories of "Mild" and "Severe" grades. The mean and standard deviation results were obtained for each group.

The statistical independent variable was age (60 years and older, and below 60 years), and the dependent variable was the mobility of the 8 joints. Differences between adults and older adults were examined in the elderly and adult cohorts of the study population. Independent samples t-test was utilized to assess the dimensional disparities between older adults and adults. SPSS 26 software was used to analyze all data, with confidence level set to 0.05.

## Result and discussion

In this study, the differences in the performance of joint movement angles between the elderly group and the adult group in the four distinct joint parts of the elbow joint, shoulder joint, trunk forward flexion, and knee joint in comfortable and effortful states were analyzed using an independent t-test research methodology. As shown in Table 2, the statistical test results demonstrated a significant difference in the performance of the elderly group and the adult group at the 0.01 level in the mild grade of the elbow joint ($p < 0.01$). In combination with the mean results, the mild grade angle of the elbow joint was 82.68˚ in the elderly group and 33.16˚ in the adult group; The statistical test results demonstrated a significant difference in the performance of the elderly group and the adult group at the 0.01 level in the mild grade of the shoulder joint ($p < 0.01$). In combination with the mean results, the mild grade angle of the shoulder joint was 119.58˚ in the elderly group and 157.85˚ in the adult group; The statistical test results demonstrated a significant difference in the performance of the elderly group and the adult group at the 0.01 level in the mild grade of the torso forward bending angle ($p < 0.01$). In combination with the mean results, the mild grade angle of the the torso forward bending angle was 31.82˚ in the elderly group and 58.19˚ in the adult group; The statistical test results demonstrated a significant difference in the performance of the elderly group and the adult group at the 0.01 level in the mild grade of the knee joint ($p < 0.01$). In combination with the mean results, the mild grade angle of the knee joint was 139.95˚ in the elderly group and 117.47˚ in the adult group; The statistical test results demonstrated a significant difference in the performance of the elderly group and the adult group at the 0.01 level in the sever grade of the elbow joint ($p < 0.01$). In combination with the mean results, the sever grade angle of the

**Table 2. The eight joint mobility measures in study subjects by age.**

| Dimensions | Group | Mean | SD | t | Sig. (2-tailed) |
|---|---|---|---|---|---|
| Elbow joint angle–Mild grade | Seniors (n = 62) | 82.68 | 21.702 | 17.209 | $p < 0.001$ |
| | Adults (n = 62) | 33.16 | 6.506 | | |
| Shoulder joint angle–Mild grade | Seniors (n = 62) | 119.58 | 19.432 | -12.914 | $p < 0.001$ |
| | Adults (n = 62) | 157.85 | 12.922 | | |
| Torso forward bending angle–Mild grade | Seniors (n = 62) | 31.82 | 5.228 | -23.1 | $p < 0.001$ |
| | Adults (n = 62) | 58.19 | 7.312 | | |
| Knee angle–Mild grade | Seniors (n = 62) | 139.95 | 9.862 | 11.719 | $p < 0.001$ |
| | Adults (n = 62) | 117.47 | 11.444 | | |
| Elbow joint angle–Sever grade | Seniors (n = 62) | 42.08 | 10.053 | 12.755 | $p < 0.001$ |
| | Adults (n = 62) | 22.47 | 6.748 | | |
| Shoulder joint angle–Sever grade | Seniors (n = 62) | 151.42 | 10.213 | -17.622 | $p < 0.001$ |
| | Adults (n = 62) | 176.52 | 4.63 | | |
| Torso forward bending angle–Sever grade | Seniors (n = 62) | 44.02 | 7.504 | -30.044 | $p < 0.001$ |
| | Adults (n = 62) | 91.82 | 10.034 | | |
| Knee angle–Sever grade | Seniors (n = 62) | 119.13 | 9.755 | 21.36 | $p < 0.001$ |
| | Adults (n = 62) | 66.21 | 16.893 | | |

elbow joint was 42.08˚ in the elderly group and 22.47˚ in the adult group; The statistical test results demonstrated a significant difference in the performance of the elderly group and the adult group at the 0.01 level in the sever grade of the shoulder joint (p<0.01). In combination with the mean results, the sever grade angle of the shoulder joint was 151.42˚ in the elderly group and 176.52˚ in the adult group; The statistical test results demonstrated a significant difference in the performance of the elderly group and the adult group at the 0.01 level in the sever grade of the torso forward bending angle (p<0.01). In combination with the mean results, the sever grade angle of the torso forward bending angle was 44.02˚ in the elderly group and 91.82˚ in the adult group; The statistical test results demonstrated a significant difference in the performance of the elderly group and the adult group at the 0.01 level in the sever grade of the knee joint (p<0.01). In combination with the mean results, the sever grade angle of the knee joint was 119.13˚ in the elderly group and 66.21˚ in the adult group. In conclusion, the anthropometric results showed highly significant differences in joint mobility in all dimensions in the elderly group compared to the adult group (p < 0.01).

The optimal reachable dimension of any space contributes to the comfort and safety of the user [28–30]. Based on the data collection results in the previous step, a model for calculating the attainable heights of the vertical dimension for elderly people is constructed, As shown in Fig 4.

Paying special attention to the fact that elderly people are reluctant to reach their limit during home activities, although, in principle, it may be good training to keep their muscles soft [15]. Therefore, the height parameter "H" does not calculate the hand dimension. The hand dimension is used as a margin for reaching the limit to accommodate this particular behavioural need of the elderly. The height parameter "H" is schematically calculated. Let the elbow joint angle be $\theta_1$, the shoulder joint angle be $\theta_2$, the torso angle be $\theta_3$, the knee angle be $\theta_4$, and the calf angle be $\theta_5$; let the forearm length be $l_1$, the upper arm length be $l_2$, the torso length be $l_3$, the thigh length be $l_4$, and the tibial point height be $l_5$. "H" can be expressed using the following equation:

$$H = l_1 \times \cos(\theta_1 + \theta_2 - \theta_3) + l_2 \times \cos(180 - \theta_2 + \theta_3) + l_3 \times \cos\theta_3 + l_4 \times \sin(\theta_4 - \theta_5) + l_5 \times \sin\theta_5$$

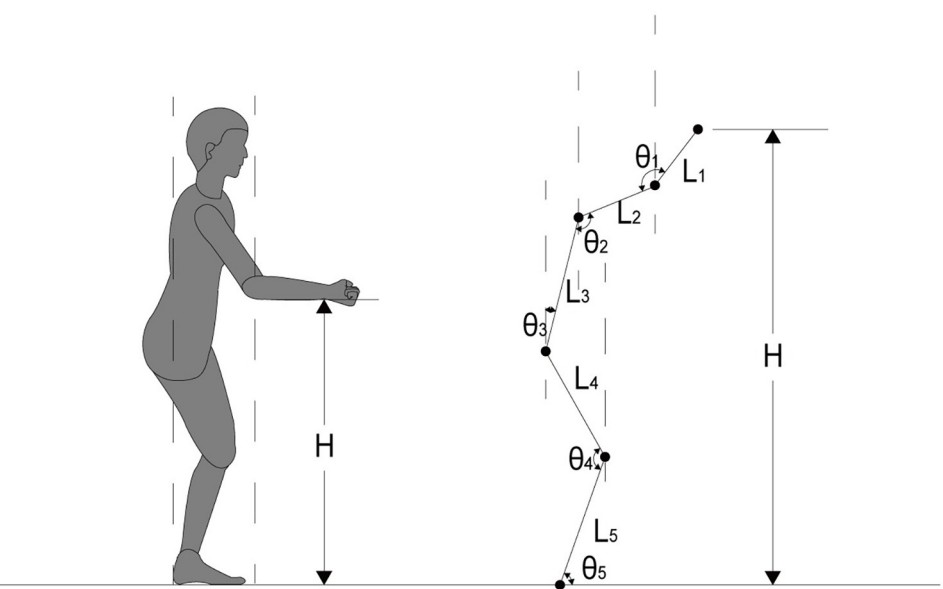

**Fig 4. The model for calculating the attainable heights of the vertical dimension.**

## Generation of each reference layer of the spatial vertical dimensional layout system

The complex and numerous types of inclusions in the interior residential space require the designer to handle the placement of all inclusions in the space from the perspective of a global system. The system should be based on the ability of the vertical contact dimension corresponding to the different standing behavioural postures of elderly users. According to the angle values corresponding to the three grades collected from each joint, each contact dimension is substituted into the vertical dimension reachable height calculation model to establish eight vertical reference layers of residential space for the contact dimension capability of elderly users. Reference layers will be generated. Also, the three static physiological dimensions of the body, namely height, eye height and shoulder height, are selected to be closely related to the vertical layout, e.g., height limits the minimum height of the vertical dimensional channel, eye height affects the height of inclusions with line of sight demand, shoulder height limits the minimum height of protrusions at the side interface of the traffic space, etc. Adding the height reference layer, the eye-height reference layer, and the shoulder height reference layer bring this total to 11 reference layers. Together, these constitute the vertical dimension layout system of indoor space for elderly users.

**The reachable layer of the shoulder joint under severe conditions (Abbreviation: Layer-SS).** Layer -SS is the highest reachable plane of the body of elderly users. As shown in Fig 5.

Using the model calculation equation, the shoulder joint angles were substituted for the "Severe" grade data, and the rest of the joint angles were substituted for the "Neutral" grade value. Layer-SS often provides a vertical height ceiling reference for relatively low-use residential contents, for example, the height of the topmost shelf.

**The reachable layer of the shoulder joint under mild conditions (Abbreviation: Layer-SM).** Layer-SM is the highest reachable plane when considering the comfort of elderly users. As shown in Fig 5.

Using the model calculation equation, the shoulder joint angles were substituted for the "mild" grade value, and the rest of the joint angles were substituted for the "Neutral" grade

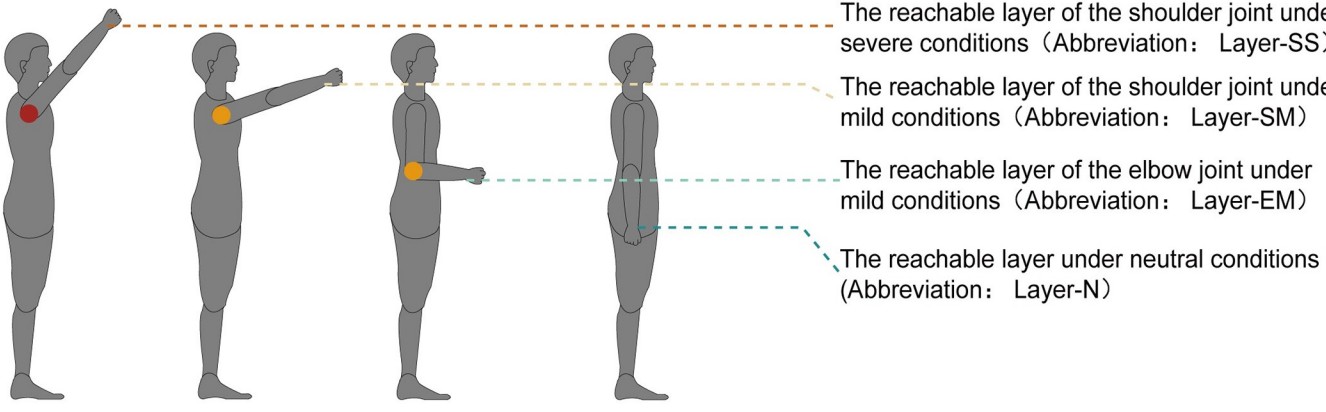

The reachable layer of the shoulder joint under severe conditions（Abbreviation：Layer-SS）

The reachable layer of the shoulder joint under mild conditions（Abbreviation：Layer-SM）

The reachable layer of the elbow joint under mild conditions（Abbreviation：Layer-EM）

The reachable layer under neutral conditions（Abbreviation：Layer-N）

**Fig 5. Four layers in the neutral state of the torso.**

value. Layer-SM often provides a vertical height ceiling reference for inclusions used relatively frequently, for example, the upper height of a towel rack.

**The reachable layer of the elbow joint has mild conditions (Abbreviation: Layer-EM).** Layer-EM is the height of the operating platform plane when elderly users are in a comfortable standing posture [24]. As shown in Fig 5, The common operating platform height for adults is around 90˚ at the elbow [24]. However, the data collection shows that some elderly people cannot flex and extend the elbow joint at 90˚ under mild conditions. The average angle of the mild-grade elderly people's elbow joint is 82. 68 ± 21. 70˚. Using the model calculation equation, if the value of the elbow joint's mild grade is less than or equal to 90˚, the height of a console for an elderly user is calculated in the same way as that for an adult, i.e., the elbow joint angle is substituted into the 90˚ value, and the opposite is substituted into the elderly elbow joint "Mild" grade value. The other joint angle values are substituted into the "Neutral" grade value. Layer-EM often provides a height reference for the more delicate daily operating surfaces, for example, the height of surfaces used to sort daily food and medicine and the height of switches, sockets, and other inclusions that the elderly use.

**The reachable layer under neutral conditions (Abbreviation: Layer-N).**　Layer-N is the lowest reachable plane when elderly users are in a natural standing state. As shown in Fig 5.

Using the model calculation equation, the values of the joint angles of each part are substituted into the "Neutral" grade values. Layer-N is often used to provide a vertical height reference for the daily use of surfaces reachable by hand when naturally standing, such as the height of support armrests, shelf hooks, and drawer handles.

## The reachable layer of the torso under mild conditions (Abbreviation: Layer-TM)

As shown in Fig 6, Layer-TM is the accessible plane with torso bending behaviour when considering the comfort of elderly users.

Using the model calculation equation, the torso angle is substituted into the "Mild" grade value. The "Severe" grade value of the shoulder joint is judged against the torso angle value. Suppose the "Severe" grade angle is smaller than the torso angle. In that case, the shoulder joint angle is substituted into the "Severe" grade value (according to the previous research results, the average shoulder joint mild grade angle is 119.58 ± 19 .43; the average angle of the mild grade of the torso forward flexion angle is 31.82 ± 5.23, which indicates that the probability of this type of occurrence is infinitely close to 0; the opposite is true for the torso angle, and

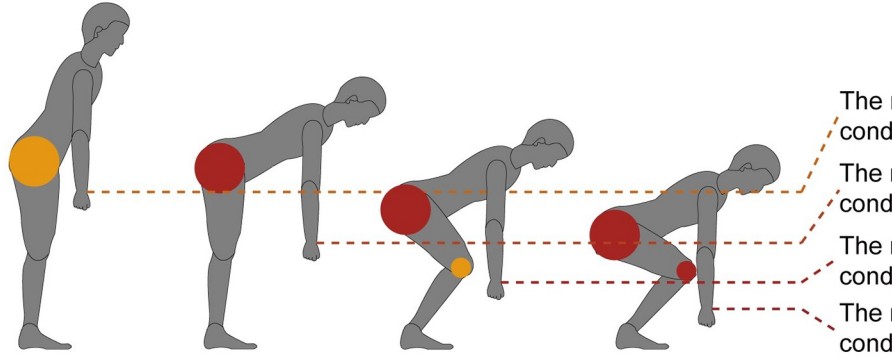

**Fig 6. Four layers in the flexion state of the torso.**

the rest of the joint angles are substituted into the "Neutral" grade. Layer-TM provides lower reference values for relatively more frequently used inclusions that require bending, such as the support surfaces of storage drawers used daily.

**The reachable layer of the torso under severe conditions (Abbreviation: Layer-TS).** Layer-TS is the lowest reachable plane with torso dips of elderly users. As shown in Fig 6.

This layer is calculated the same way as Layer-TM, with the torso angle and shoulder joint angle substituted into the "Severe" grade of the torso and the rest of the joint angles substituted into the "Neutral" grade. Layer-TS often provides the minimum lower reference for inclusions that require bending, for example, the lowest interface for placing dishes in the dishwasher.

**The reachable layer of the knee under mild conditions (Abbreviation: Layer-KM).** Layer-KM provides the lowest reachable reference for the relatively mild knee flexion movement of elderly users. As shown in Fig 6.

The elderly user must perform torsi forward and knee forward flexion to reach a plane of relative comfort. Using the model calculation equation, the torso angle and shoulder angle are substituted into the "Severe" grade value for the torso, the "Mild" grade value for the knee, and the"Neutral" grade value for the elbow—for example, the compartment grade of a low shelf. Layer-KM often provides a surface reference for underutilised content, for example, the height of those shelf dividers used to store seasonal clothing.

**The reachable layer of the knee under severe conditions (Abbreviation: Layer-KS).** Layer-KS provides the lowest reachable reference for elderly users, as is shown in Fig 6. The elderly user must perform Torso forward and knee forward flexion in the lowest reachable plane. Using the model calculation equation, the torso angle and shoulder angle are substituted into the "Severe" grade values for the torso, the"Severe" grade data for the knee, and the"Neutral" grade values for the elbow angle. Layer-KS often provides a vertical height lower limit reference for elderly users, for example, the height of the bottom interface of lockers.

**Other reference layers.**  Three important reference layers can be directly obtained via measurements of the neutral state condition. The height reference layer (abbreviation: Layer-H) often provides a lower height reference for the passage and hanging inclusions above the human head, for example, the height of a door. The eye-height reference layer (abbreviation: Layer-EH) often provides a height reference for inclusions with the line-of-sight requirements, for example, mirror height, clock height, and screen partition height. The shoulder height reference layer (abbreviation: Layer-SH) often provides a height reference for the shoulders of people in traffic spaces that may touch the lower limits of the inclusions, for example, the height of partitions against walls and the heights of movable showers.

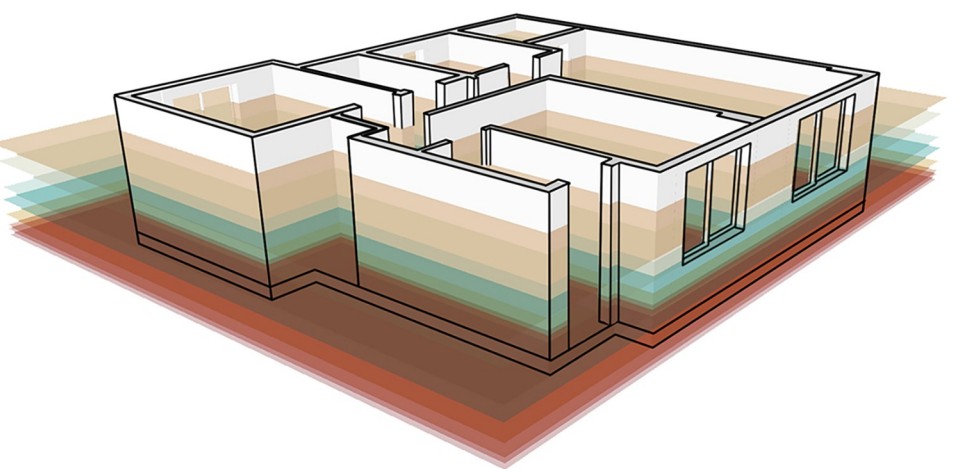

**Fig 7. Residential space is divided vertically into 11 reference layers.**

## Application of vertical space layout system

A residential indoor space environment is a system that includes all the inclusions in a space in a certain order to form an organic complex. Together, these inclusions provide environmental support for the home activities of elderly users. This study vertically divides indoor space into 11 reference layers according to the physical activity ability of elderly users. As shown in Fig 7, each layer is interconnected, and together, they define the vertical space layout of the elderly users' residential interiors.

Studies have shown that 21 per cent of people over 65 and 55 per cent of people over 85 frequently experience behavioural posture disorders in their daily lives [31]. Non-neutral postures, such as squatting and torso flexion, are strongly associated with lower back pain in the elderly [32]. Therefore, in designing the vertical space layouts of residential interiors, attempts should be made to aid the maintenance of a natural body posture and reduce unnecessary bending and stretching postures [33]. According to this principle, combined with the posture assessment criteria determined by each layer, this vertical division system is divided into six zones according to the degree of safety and comfort, of which the most comfortable are in zone I. As shown in Fig 8. Place as many inclusions as possible in areas of high comfort as conditions permit.

Washing spaces can be challenging for older adults living independently [34]. This study shows the application method of each reference surface within this system, as shown in Fig 9, using a vanity space arrangement as an example. The washing space mainly contains a shower

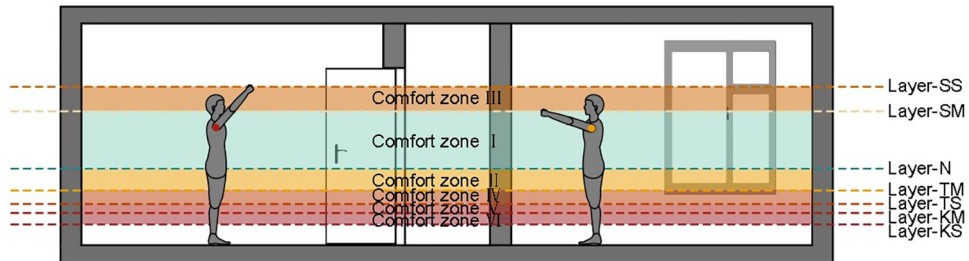

**Fig 8. Comfort zones.**

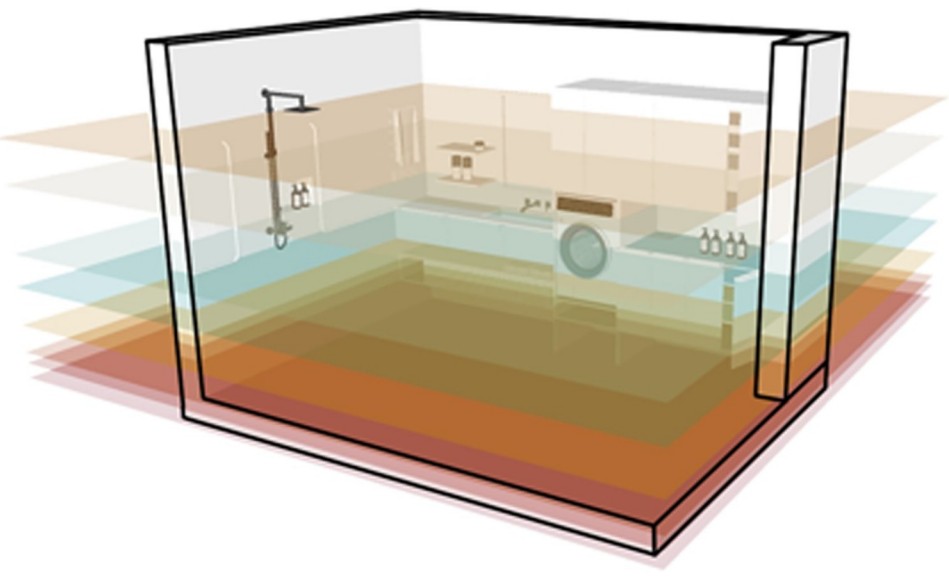

**Fig 9. Vertical division of the washing space.**

area, a sink area, a laundry area, and a storage area. There are many types and numbers of inclusions in this space. The vertical space layout for this area is designed as shown in Fig 9. The general layout design policy is that all inclusions are located in the area between Layer-SS and Layer-KS.

The vertical space layout of the shower area is designed as shown in Fig 10, and the number of residential inclusions in this area is relatively small. Combined with the actual use of the area inclusions concentrated in a comfort zone I, the vertical layout is mainly determined by Layer-SS, Layer-SM, Layer-EM, and Layer-N together.

A shower nozzle consists of two fixed and movable nozzles, and the inclusions are vertical because they are located in the main traffic area and are part of the shape of the projection of the wall. To avoid causing excessive obstruction to the main traffic, the first should meet the lower limit requirements of the height of Layer-SH; at the same time, the inclusions are required to spray the whole body, so at least the vertical position of the fixed nozzle should also consider the lower limit requirements of the height of Layer-H. Finally, combined with the joint mobility of elderly users, the fixed shower head is located above Layer-H, the movable

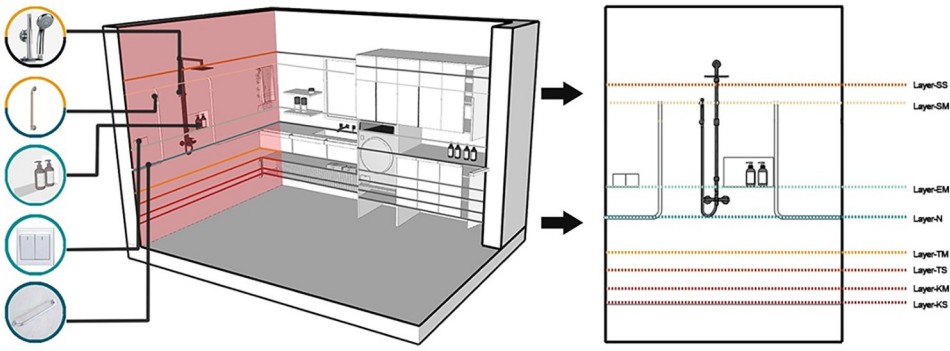

**Fig 10. Vertical dimensional division of space in the shower area.**

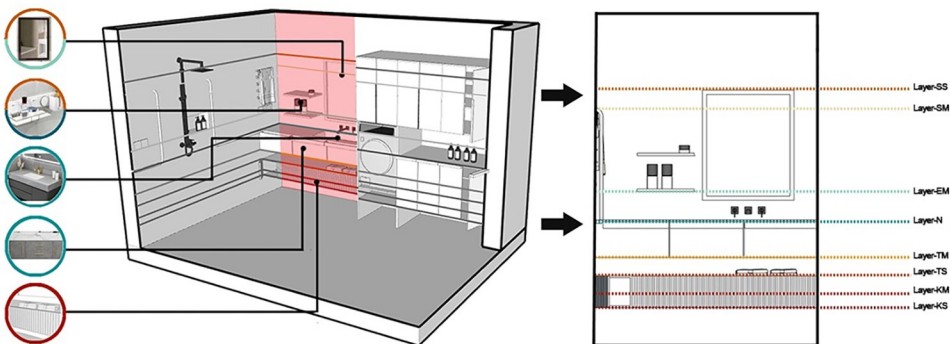

**Fig 11. Vertical dimensional division of space in the washbasin area.**

shower head is placed between Layer-SM and Layer-N, and the nozzle switch is often placed in Layer-N. To aid the balance of elderly users in the shower, handrails should be placed vertically because vertical handrails are convenient for elderly users. The corresponding area for this is between Layer-SM and Layer-N; the vertical position of the switch and toiletry shelf used by the elderly is in Layer-EM.

The vertical space layout of the washbasin area is designed as shown in Fig 11, and the number of residential inclusions in this area is relatively large. Combined with the actual use of the vertical layout of the distribution of comfort zones Izone, all reference layers must jointly determine me is zone I, II, IV, V and VI in this area. The high position of the shelf contains the shelf platform and towel rack, which are used relatively frequently, so it is located in the comfort zone, that is, between Layer-SM and Layer-N. The placement of the mirror should meet the requirements of Layer-EH. To assist elderly users in maintaining balance when walking, a horizontal handrail is placed. The horizontal handrail is combined with the cabinet switch handle, with the vertical height in Layer N. The low storage cabinet is located in the area between Layer-N and Layer-KS. More frequently used inclusions should be placed in comfort zone II, and less frequently used inclusions should be placed in low-comfort areas.

The vertical space layout of the laundry and storage area is designed as shown in Fig 12, The number of residential inclusions in this area is relatively small, but the volume of the inclusions is larger, and they are spread all over comfort zones to VI. The upper limit of the locker shelves with a high position is located in Layer-SS, and the items with a high frequency of use are placed in Layer-SM as far as possible inside the locker. Combined with the items'

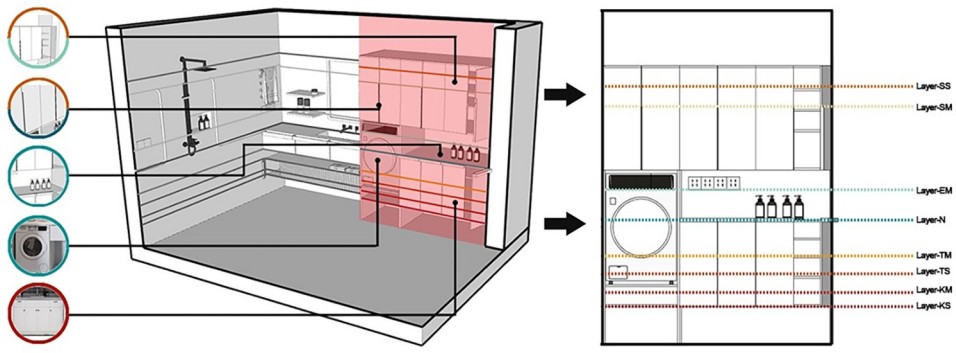

**Fig 12. Vertical division of the laundry and storage area.**

volume, the partition's vertical layout is decided, and the upper limit of the door armrest of the cabinet in this area is also arranged in Layer-SM. The middle area has a higher frequency of daily use and is arranged in the form of an open storage compartment, and it is located between Layer EM and Layer N. The low position of the locker with the sink area and the internal shelves can be placed in the comfort zone layer. The location of the washing machine and the location of the washing machine bucket used to pick up clothes should at least be in comfort zone VI, that is, Layer-KS and the area should be elevated as much as possible in comfort zones I and II; at the same time, the washing machine operation panel location should be in the view of the upper limit of the door panel. The operating panel should be below the line of sight, i.e., below Layer-EH. Ultimately, multiple reference layers are used to determine the vertical layout of the washing machine.

## Limitations

The anthropometric data measurements in this study require professional laboratory personnel. However, it takes time, and effort to train professional laboratory personnel, and measurements taken by non-professional laboratory personnel are prone to large errors. Meanwhile, the method focuses on the standing posture, the most frequently used and dangerous posture, while other behavioural postures are not overly concerned for the time being. Finally, this study collects anthropometric data from a sample of elderly people in Shandong Province, China; the coverage of the study sample is somewhat limited, so the results generated in this study are not universal at present.

## Conclusion

The significant decline in the joint mobility of elderly people harms their independent living. In this study, the accessibility range of the vertical dimension of elderly people is used as a criterion to measure the decline in their joint mobility, and 13 variables in the anthropometric dimension are selected for measurement and data collection to construct a calculation model of the vertical dimension accessibility of elderly people, which provides a calculation method for the layout of the vertical dimension of elderly-friendly space. The method provides a vertical dimensional height reference for all inclusions in indoor spaces to provide a safe and comfortable home living environment for the elderly.

Further research will be conducted in three directions in the future. On the one hand, efforts will be made to find a relatively simple measurement method to improve the measurement efficiency and accuracy; on the other hand, joint angle measurement will be conducted for non-standing postures to expand the application of this method; Finally, efforts will be made to increase the number of measurement samples and to substitute the collected data into the computational model proposed in this thesis. Through the statistical analysis of the calculation results, the final reference of each reference floor elevation for the vertical dimension layout of indoor space applicable to most of the elderly population is proposed.

## Supporting information

**S1 File.**
(XLSX)

## Acknowledgments

This study is especially grateful to and derived from the PhD program in design at Silpakorn University's School of Decorative Arts.

## Author Contributions

**Formal analysis:** Jie Liu.

**Investigation:** Bo Zhang.

**Supervision:** Isarachai Buranaut.

**Writing – original draft:** Feng Wang.

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
