## [Decision Letter · Decision Letter 0]

20 Feb 2023

PONE-D-22-33201

Elderly-friendly indoor Vertical Dimensional Layout Method based on Joint Mobility

PLOS ONE

Dear Dr. Liu,

Thank you for submitting your manuscript to PLOS ONE. After careful consideration, we feel that it has merit but does not fully meet PLOS ONE’s publication criteria as it currently stands. Therefore, we invite you to submit a revised version of the manuscript that addresses the points raised during the review process.

We look forward to receiving your revised manuscript.

Kind regards,

Monika Błaszczyszyn

Academic Editor

PLOS ONE

 Journal Requirements:

2.Please ensure that you have specified (1) whether consent was informed and (2) what type you obtained (for instance, written or verbal, and if verbal, how it was documented and witnessed). If your study included minors, state whether you obtained consent from parents or guardians. If the need for consent was waived by the ethics committee, please include this information.

Reviewers' comments:

Reviewer's Responses to Questions

**Comments to the Author**

1. Is the manuscript technically sound, and do the data support the conclusions?

Reviewer #1: No

Reviewer #2: Partly

2. Has the statistical analysis been performed appropriately and rigorously? 

Reviewer #1: No

Reviewer #2: No

3. Have the authors made all data underlying the findings in their manuscript fully available?

Reviewer #1: No

Reviewer #2: Yes

4. Is the manuscript presented in an intelligible fashion and written in standard English?

Reviewer #1: Yes

Reviewer #2: Yes

5. Review Comments to the Author

Reviewer #1: I reviewed the manuscript titled ' Elderly-friendly indoor Vertical Dimensional Layout Method based on Joint Mobility', there are some remarks that should be considered:

1- There are many grammatical mistakes and typos that should be revised carefully.

2- The abstract is not structured, and should be following subheadings, like introduction, etc.

3- The importance of the study is not described in the abstract.

4- The statistical analysis is not well reported, moreover, the statistical software is not mentioned.

5- The most important point of the findings is not well-emphasized. What was the final finding and novelty of the study? It should be discussed.

6- How did the authors estimate the sample sized and haw was the sampling method.

7- The baseline and demographic data of groups are not mentioned.

8- Obviously, the orthopedic status of the elderly would be worse than the younger group, what was the novel findings of the study? They need to be reported and discussed.

Reviewer #2: Substantive comments to the Authors:

The subject matter addressed is certainly relevant. The increasing population of people in elderly and people in general with incomplete range of motion is a challenge for designers. The research presented in the article characterizes selected aspects related to comfortable access to usable space. I would expect the authors to indicate the novelty of the presented approaches:

- in the area of measurement methodology,

- measurement techniques,

- the scope of the research (e.g., a significant increase in the study group).

In its current form, the presented content does not indicate significant progress in the indicated areas. It would also be appropriate to supplement the presented results with a statistical analysis of the reliability of the tested hypotheses.

The following substantive issues also need clarification:

1."This study vertically divides indoor space into 11 reference layers according to the physical activity ability of the elderly users" -> in the text, I found no justification for such a division. Where does this number come from?

2. Line 402 ... apply this research method ... -> What exactly the authors mean by the term "this research method"?

I hope the above comments will improve this interesting article.

Editing notes:

Line:

68 Sixty -two -> Sixty-two

80 in ISO 7250—basic human -> in ISO 7250 — basic human

150 action for 10 -15s -> this provision is ambiguous

170 and Fig 3 -10. -> and Fig 3-10.

274 As shown in Fig 9. -> this sentence seems to be unfinished

316 postures[30]. - > postures [30].

Fig 3-6 -> descriptions on axes too small and unreadable

Fig 13-14 -> descriptions are not very readable

6. PLOS authors have the option to publish the peer review history of their article (what does this mean?). If published, this will include your full peer review and any attached files.

Reviewer #1: No

Reviewer #2: No

---

## [Author Response · Author response to Decision Letter 0]

6 Apr 2023

Dear Monika Błaszczyszyn,

Thank you for giving us the opportunity to submit a revised draft of the manuscript “Elderly-friendly indoor Vertical Dimensional Layout Method based on Joint Mobility” for publication in the Journal of PLOS ONE. We appreciate the time and effort that you and the reviewers dedicated to providing feedback on our manuscript and are grateful for the insightful comments on and valuable improvements to our paper. 

We have revised the manuscript and answered reviewer's questions/comments with point by point hereunder according to the requirements in your letter. Hence, we are re-submitting herewith the revised manuscript.

---

## [Decision Letter · Decision Letter 1]

2 May 2023

Elderly-friendly indoor Vertical Dimensional Layout Method based on Joint Mobility

PONE-D-22-33201R1

Dear Dr. Liu,

We’re pleased to inform you that your manuscript has been judged scientifically suitable for publication and will be formally accepted for publication once it meets all outstanding technical requirements.

Kind regards,

Monika Błaszczyszyn

Academic Editor

PLOS ONE

Additional Editor Comments (optional):

Reviewers' comments:

Reviewer's Responses to Questions

**Comments to the Author**

1. If the authors have adequately addressed your comments raised in a previous round of review and you feel that this manuscript is now acceptable for publication, you may indicate that here to bypass the “Comments to the Author” section, enter your conflict of interest statement in the “Confidential to Editor” section, and submit your "Accept" recommendation.

Reviewer #1: All comments have been addressed

Reviewer #2: All comments have been addressed

2. Is the manuscript technically sound, and do the data support the conclusions?

Reviewer #1: Yes

Reviewer #2: Yes

3. Has the statistical analysis been performed appropriately and rigorously? 

Reviewer #1: Yes

Reviewer #2: Yes

4. Have the authors made all data underlying the findings in their manuscript fully available?

Reviewer #1: No

Reviewer #2: Yes

5. Is the manuscript presented in an intelligible fashion and written in standard English?

Reviewer #1: Yes

Reviewer #2: Yes

6. Review Comments to the Author

Reviewer #1: All of my comments have been addressed and manuscript sounds to be appropriate for publication in the present status. Thank you

Reviewer #2: The amendments made to the text have significantly improved the readability of the content conveyed.

7. PLOS authors have the option to publish the peer review history of their article (what does this mean?). If published, this will include your full peer review and any attached files.

Reviewer #1: No

Reviewer #2: No

---

## [Editor Report · Acceptance letter]

10 May 2023

PONE-D-22-33201R1 

Elderly-friendly indoor Vertical Dimensional Layout Method based on Joint Mobility 

Dear Dr. Liu:

I'm pleased to inform you that your manuscript has been deemed suitable for publication in PLOS ONE. Congratulations! Your manuscript is now with our production department. 

Kind regards, 

on behalf of

Dr. Monika Błaszczyszyn 

Academic Editor

PLOS ONE